# Pseudotyped Bat Coronavirus RaTG13 is efficiently neutralised by convalescent sera from SARS-CoV-2 infected patients

Diego Cantoni[1], Martin Mayora-Neto [1], Nazia Thakur [2,3], Ahmed M. E. Elrefaey [2], Joseph Newman [2], Sneha Vishwanath[4], Angalee Nadesalingam[5], Andrew Chan[5], Peter Smith[5], Javier Castillo-Olivares[5], Helen Baxendale[6], Bryan Charleston[2], Jonathan Heeney [4,5], Dalan Bailey [2✉] & Nigel Temperton [1✉]

RaTG13 is a close relative of SARS-CoV-2, the virus responsible for the COVID-19 pandemic, sharing 96% sequence similarity at the genome-wide level. The spike receptor binding domain (RBD) of RaTG13 contains a number of amino acid substitutions when compared to SARS-CoV-2, likely impacting affinity for the ACE2 receptor. Antigenic differences between the viruses are less well understood, especially whether RaTG13 spike can be efficiently neutralised by antibodies generated from infection with, or vaccination against, SARS-CoV-2. Using RaTG13 and SARS-CoV-2 pseudotypes we compared neutralisation using convalescent sera from previously infected patients or vaccinated healthcare workers. Surprisingly, our results revealed that RaTG13 was more efficiently neutralised than SARS-CoV-2. In addition, neutralisation assays using spike mutants harbouring single and combinatorial amino acid substitutions within the RBD demonstrated that both spike proteins can tolerate multiple changes without dramatically reducing neutralisation. Moreover, introducing the 484 K mutation into RaTG13 resulted in increased neutralisation, in contrast to the same mutation in SARS-CoV-2 (E484K). This is despite E484K having a well-documented role in immune evasion in variants of concern (VOC) such as B.1.351 (Beta). These results indicate that the future spill-over of RaTG13 and/or related sarbecoviruses could be mitigated using current SARS-CoV-2-based vaccination strategies.

[1] Viral Pseudotype Unit, Medway School of Pharmacy, Universities of Kent & Greenwich, Chatham, UK. [2] The Pirbright Institute, Guildford, Surrey GU24 0NF, UK. [3] The Jenner Institute, Nuffield Department of Medicine, University of Oxford, Oxford, UK. [4] Laboratory of Viral Zoonotics, Department of Veterinary Medicine, University of Cambridge, Cambridge, UK. [5] DIOSynVax, University of Cambridge, Madingley Road, CB3-0ES Cambridge, UK. [6] Royal Papworth Hospital NHS Foundation Trust, Cambridge, UK. ✉email: dalan.bailey@pirbright.ac.uk; n.temperton@kent.ac.uk

The Coronavirus disease 2019 (COVID-19) pandemic, caused by the severe acute respiratory syndrome coronavirus 2 (SARS-CoV-2), has currently surpassed 270 million recorded cases, claimed upwards of 5 million lives and continues to overwhelm healthcare facilities in countries around the world[1,2]. SARS-CoV-2 is a betacoronavirus, as are the close relative SARS-CoV and Middle Eastern respiratory syndrome-related virus (MERS), both of which have spilled over into human populations from animal reservoirs. The natural reservoir of alpha and betacoronaviruses is widely believed to be bats. The direct ancestor of SARS-CoV-2 remains to be identified, as well as a possible intermediate reservoir. More than 250 coronaviruses have been detected in bats[3].

In 2013, a bat coronavirus was detected in Mojiang County, Yunnan, China, eventually denoted as RaTG13. Following the emergence of SARS-CoV-2 in 2019, the RaTG13 isolate was shown to be one of the closest known relatives, with 96.2% sequence similarity at the genome-wide level[1]. Of note, isolates of related viruses in pangolins have receptor binding domains (RBDs) which are more closely related, likely a broader reflection of frequent sarbecovirus recombination in reservoirs[4]. To date, the direct ancestor of SARS-CoV-2 remains to be identified, as well as any possible intermediate reservoir. Nevertheless, SARS-CoV-2 and RaTG13 both rely on their Spike protein for viral entry into cells, via the cell surface angiotensin-converting enzyme 2 (ACE2) receptor, complemented by activity of additional proteases including Transmembrane protease, serine 2 (TMPRSS2). Comparative studies of the structures of both SARS-CoV-2 and RaTG13 Spike revealed a high degree of conservation in the ectodomain (97.8%) with most of the substitutions being within the RBD, clustering around the ACE2-binding interface and the RBD-RBD interfaces of the trimeric Spike complex[5,6]. Previously, we have identified that introducing SARS-CoV-2 changes into RaTG13 Spike increases the tropism for human ACE2, providing mechanistic insight into the potential pathway for spill-over[7]. Of note, these mutants were constructed in Spike expression constructs for pseudotyping and were not inserted into recombinant RaTG13 or SARS-CoV-2 viruses. Given the structural similarities between SARS-CoV-2 and RaTG13 Spike, as well as the likelihood of a shared ancestor, we wanted to assess whether antibodies generated following prior infection with SARS-CoV-2 or through Spike-protein-based vaccination, would result in neutralisation of RaTG13. This is particularly relevant in the context of emerging variants of concern (VOCs), e.g. B.1.1.7 (Alpha), B.1.351 (Beta), B.1.617.2 (Delta) and BA.1 (Omicron) as many of the amino acid substitutions between RaTG13 and SARS-CoV-2 Spike are at positions that have been established as important (antigenically or functionally) in VOCs, e.g. 484 and 501. It is essential we develop an understanding of the breadth of immunity conferred by both infection and vaccination. Indeed, knowledge on whether closely related coronaviruses to SARS-CoV-2 are neutralised by current vaccination programs can be used to gauge the risk of potential sarbecovirus spillover events in the future.

To characterise the degree of cross-neutralisation between these related Spikes, we pseudotyped SARS-CoV-2 and RaTG13 using a lentiviral core and performed pseudovirus microneutralisation assays (pMN) to assess the neutralisation potency of convalescent sera derived from previously infected SARS-CoV-2 patients and vaccinated healthcare workers (HCWs) (sampled in the United Kingdom). Surprisingly, despite the substantial number of RBD substitutions in RaTG13 these pseudotypes were more efficiently neutralised by our sera. These results have important implications when anticipating further zoonotic transmission of coronaviruses and the likely protection afforded by current vaccination approaches and immune responses to SARS-CoV-2.

## Results

To assess differences in neutralisation between SARS-CoV-2 and RaTG13, we initially used the WHO International Reference Panel for anti-SARS-CoV-2 immunoglobulin (NIBSC; 20/268) which provides five sera, one negative, and four positive, categorised by increasing amounts of neutralising antibodies specific to SARS-CoV-2. Interestingly, we observed higher neutralising potency against RaTG13, when compared to SARS-CoV-2 in three of the four sera samples, not including the negative sera control (Fig. 1a). Switching to a larger cohort and repeating the assay with 25 convalescent sera samples derived from patients and healthcare workers (HCWs) who were infected with SARS-CoV-2 during the first wave in the United Kingdom, we observed a similar trend, with RaTG13 being more efficiently neutralised when compared to SARS-CoV-2 (2.0 fold change, $p = <0.0001$) (Fig. 1b). In comparison, B.1.351 (Beta), a VOC first detected in South Africa, showed a significant reduction in neutralisation compared to SARS-CoV-2 (4.0 fold change, $p = <0.0001$), confirming previous observations[8–10]. Interestingly, the more distantly related sarbecoviruses SARS-CoV-1 and WIV16 were less efficiently neutralised than SARS-CoV-2, RaTG13 and Beta, likely reflecting distinct antigenicity (Fig. 1b). All pseudotypes were run with a panel of negative sera to ascertain background levels of neutralisation, with IC50 titres calculated through comparison with no sera controls (see Supplementary Fig 1a–e for exemplar data).

To compare these results to vaccine-derived immunity, we examined sera from HCWs who had received a first dose of either the BNT162b2 (Pfizer-BioNTech, n = 12) or AZD1222 (Oxford-AstraZeneca, $n = 9$) vaccine against SARS-CoV-2. BNT162b2 (Pfizer-BioNTech) is a lipid nanoparticle–formulated, nucleoside-modified RNA vaccine encoding prefusion stabilised, SARS-CoV-2 spike while ChAdOx1 nCoV-19 (AZD1222, Oxford-AstraZeneca) is an adenoviral-vectored vaccine expressing wild type (non-stabilised) spike. pMN assays were carried out on post-vaccination sera samples ($n = 21$), which again revealed higher neutralisation titres against RaTG13 compared to SARS-CoV-2 (1.2 fold change, $p = 0.0016$) (Fig. 1c). This difference was recapitulated when stratifying the group based on the absence of prior infection ($n = 11$) (4.5 fold change, $p = 0.001$) (Fig. 1d). Interestingly, the difference in medians between SARS-CoV-2 and RaTG13 in HCWs with documented prior infection ($n = 10$) was not significant (1.2 fold change, $p = 0.084$) (Fig. 1d), indicating that the boosted titre acquired from natural infection narrows the gap in neutralisation between SARS-CoV-2 and RaTG13. Lastly, we observed no significant difference between different vaccine types against the same pseudotyped virus in sera samples with no history of prior infection (Supplementary Fig 2). Together, these data provide compelling evidence that vaccination or natural infection provides cross-protective immunity to RaTG13, at least at the level of neutralising antibodies.

To examine the differences in viral neutralisation in more detail we utilised two mutant Spike plasmids in the SARS-CoV-2 or RaTG13 backbone[7]. These two chimeric Spikes (SARS-CoV-2 Multi RBD: N439K, Y449F, E484T, F486L, Q493Y, Q498Y, N501D, Y505H, and RaTG13 Multi RBD: K439N, F449Y, T484E, L486F, Y493Q, Y498Q, D501N, H505Y) contain the amino acid substitutions between the two viruses that are present within the RBD and known to interact with human ACE2[5,11] (Fig. 2a, b). A number of these residues are implicated in the evasion of neutralisation, either with the same amino acid change, e.g., N439K[12] or different, e.g., N501Y and E484K are seen in B.1.351[13], as opposed to N501D and E484T found in RaTG13. Analysing the same patient and HCW convalescent sera set ($n = 25$; Fig. 1), we identified that the SARS-CoV-2 Multi RBD was neutralised more efficiently than SARS-CoV-2 WT (1.9 fold change, $p = 0.0005$).

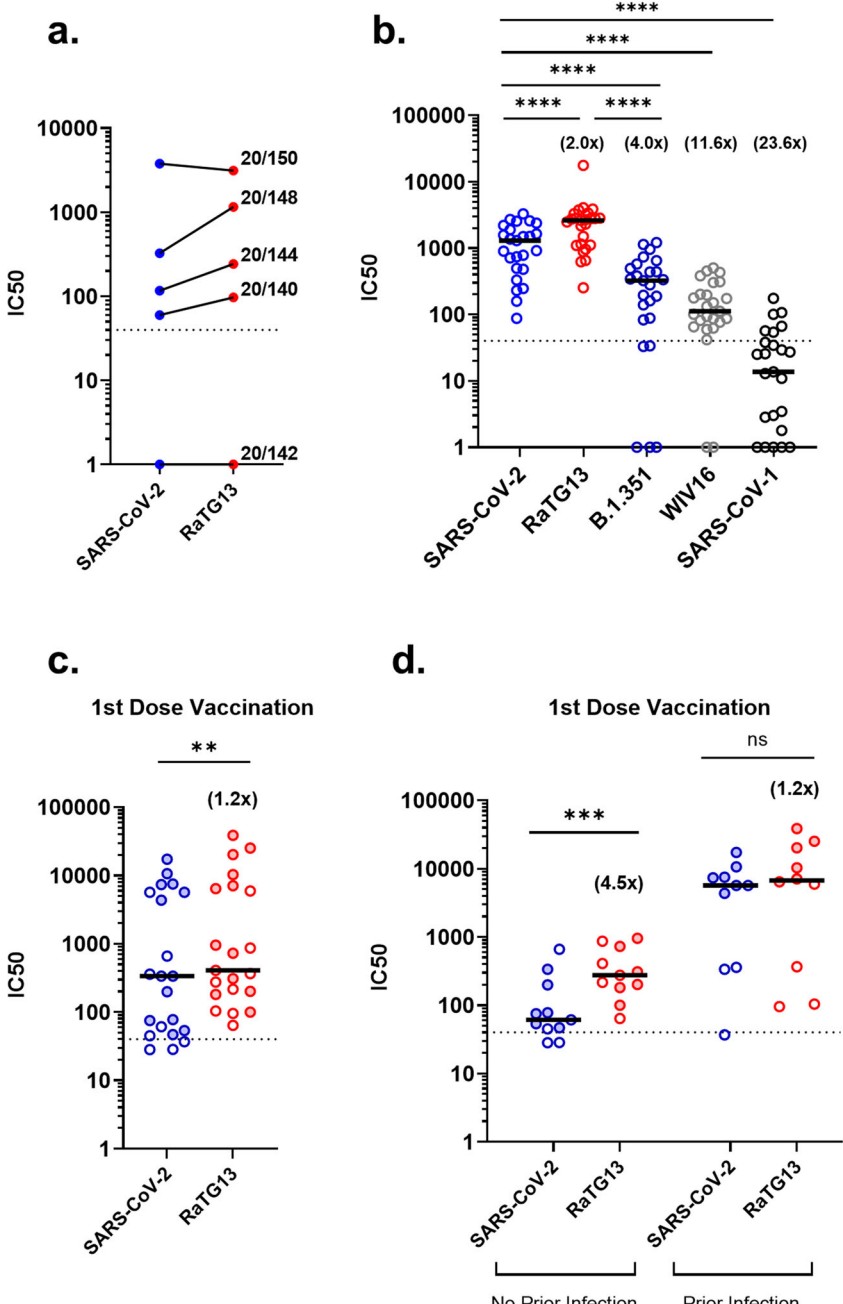

**Fig. 1 Differences in neutralisation titres between SARS-CoV-2 and RaTG13 by pMN Assay. a** Comparison of neutralisation titres between SARS-CoV-2 and RaTG13 using the WHO International Reference Panel for anti-SARS-CoV-2 immunoglobulin. Three of the four sera showed increased neutralisation titres against RaTG13. **b** Comparison of neutralisation titres between SARS-CoV-2 ($n = 25$), RaTG13 ($n = 25$, $p = <0.0001$), B.1.351 (Beta) ($n = 25$, $p = <0.0001$) variant of concern, SARS-CoV-1 ($n = 25$, $p = <0.0001$) and WIV16 ($n = 25$, $p = <0.0001$) using convalescent sera derived from patients and healthcare workers. **c** Comparison of neutralisation titres between SARS-CoV-2 and RaTG13 against sera from single-dose vaccinated healthcare workers ($n = 21$, $p = 0.0016$). **d** Differences in neutralisation titre from single dose vaccinated healthcare workers split by 'no prior infection' ($n = 11$, $p = 0.001$) or 'prior infection' ($n = 10$, $p = 0.084$) with SARS-CoV-2. Full circles denote healthcare workers vaccinated with BNT162b2 ($n = 12$), whereas open circles denote vaccination with AZD1222 ($n = 9$). Numbers in brackets denote fold changes relative to SARS-CoV-2. Wilcoxon matched pairs signed rank tests were used in panels (**b**), (**c**) and (**d**). Dotted lines in graphs denote the assay's lower limit of detection. IC50 was calculated by fitting a non-linear regression curve using Graphpad Prism 8 software. All *n* values constitute of biologically independent samples.

In contrast, the RaTG13 Multi RBD was neutralised slightly less efficiently than RaTG13 WT (1.2 fold change, $p = 0.0043$) (Fig. 2c). These relative changes in neutralisation efficiency indicated that differences in neutralisation between SARS-CoV-2 and RaTG13 (WTs) might therefore be attributable, in part, to amino acid substitutions within the Multi RBD mutants.

To identify the specific changes responsible for RaTG13's enhanced neutralisation we next examined each substitution in isolation (Fig. 2d, e), comparing their neutralisation to WT virus with four randomly chosen patient sera samples. For RaTG13 we found that the substitution of individual amino acids had little appreciable effect on neutralisation, with the

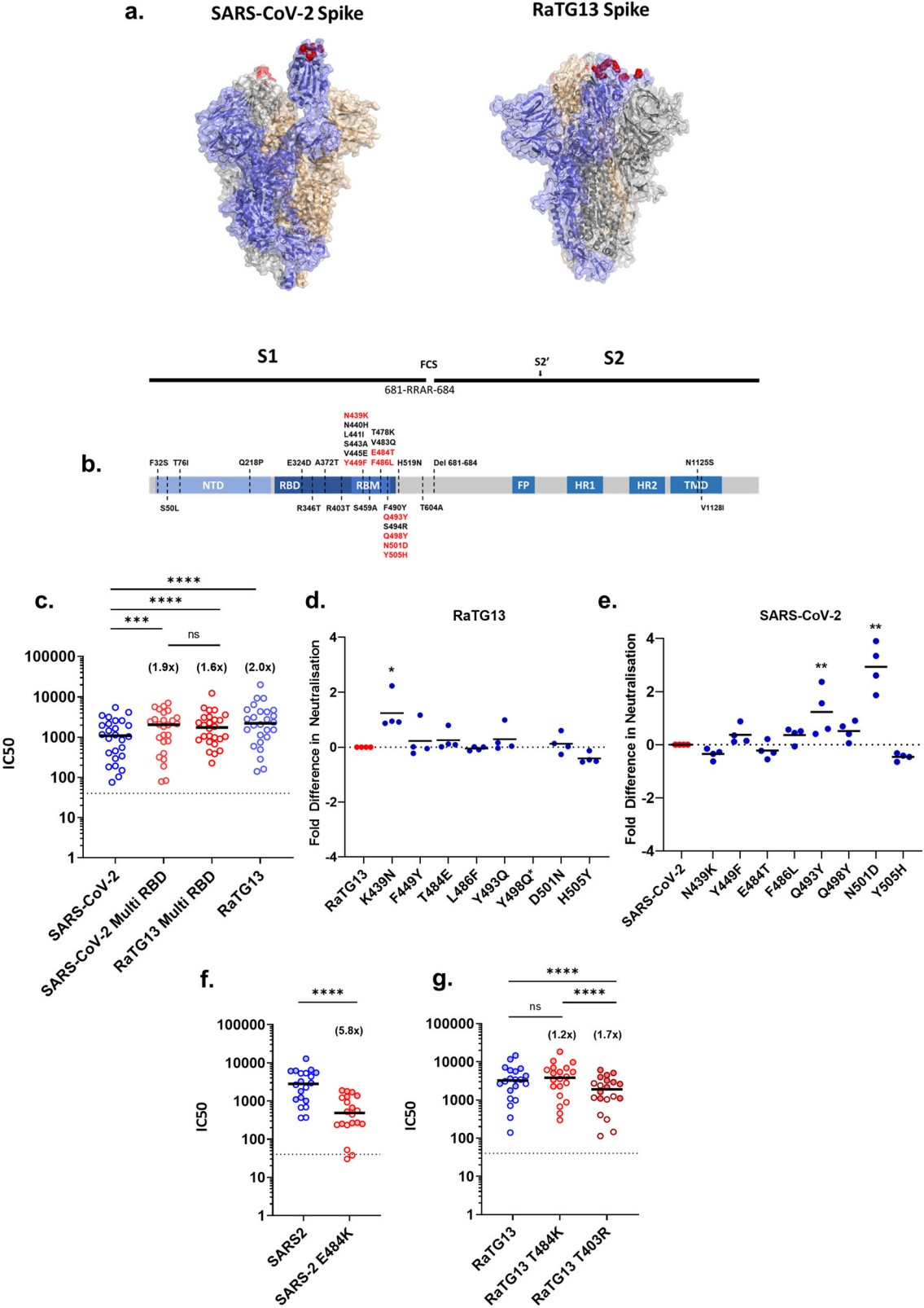

exception of K439N, which resulted in a mean 1.2 fold increase in neutralisation ($p = 0.0194$) (Fig. 2d). In contrast, several individual substitutions in the SARS-CoV-2 background (Fig. 2e) showed significant increased neutralisation potency, with Q493Y showing a mean 1.2 fold increase ($p = 0.0088$) and N501D a mean 2.9 fold increase ($p = 0.0069$). These results highlight the importance of these particular residues in defining

the relative neutralisation of RaTG13 and SARS-CoV-2 by convalescent sera. RaTG13 Y498Q neutralisations were not performed due to insufficient viral titres being attained during pseudotype preparation. Of note, all IC50s are calculated with comparison to the 'no sera' RLU recorded for the individual virus being assayed. For both the 'Multi RBD' and individual point mutants, we assessed the relative infectivity of pseudoviral

**Fig. 2 Key amino acid residues affect antibody neutralisation against SARS-CoV-2 and RaTG13. a** Structures of SARS-CoV-2 (PDB: 6X2A) and RaTG13 (PDB: 7CN4). The highlighted (red) amino acids denote the ACE2 contact residues that were substituted to generate the Multi RBD plasmids for subsequent pMN assays. **b** Simplified schematic highlighting the functional domains within the SARS-CoV-2 Spike protein. The amino acid substitutions found in RaTG13 have been labelled across the diagram, with red text denoting the substitutions displayed in panel (**a**), which were built into chimeric Spikes. (FCS, furin cleavage site; NTD, N-terminal domain; RBD, receptor binding domain; RBM, receptor binding motif; FP, fusion peptide; HR, heptad repeat; TM, transmembrane domain). (**c**) Repeated experiments showing the neutralisation titres, by pMN assay, of SARS-CoV-2 ($n = 25$), RaTG13 ($n = 25$, $p = <0.0001$) and both Sars-CoV-2 Multi RBD ($n = 25$, $p = 0.0005$) and RaTG13 Multi RBD ($n = 25$, $p = 0.0043$) mutants with the same set of convalescent sera used in Fig. 1. We did not observe a significant difference between the Multi RBD results ($n = 25$, $p = 0.9007$). **d, e** Using a set of four convalescent sera derived from patients, each pseudotype mutant was assayed by pMN and IC50s were then converted to fold changes against their original RaTG13 (**d**) (K439N; $n = 4$, $p = 0.0194$) or SARS-CoV-2 (**e**) (Q439Y; $n = 4$, $p = 0.0088$, N501D; $n = 4$, $p = 0.0069$) background. *RaTG13 Y498Q was not performed due to low PV titre (Supplementary Fig 3). **f** Lysine substitution at position 484 in SARS-CoV-2 Spike significantly reduced neutralisation ($n = 20$, $p = <0.0001$). **g** Lysine substitution at position 484 in RaTG13 however showed a subtle, non-significant increase in neutralisation titres ($n = 20$, $p = 0.1054$), whereas an arginine substitution at position 403 significantly reduced the neutralisation titres compared to RaTG13 ($n = 20$, $p < 0.0001$) and RaTG13 T484K ($n = 20$, $p < 0.0001$). Comparisons of neutralisation in panel (**f**) and (**g**) were made using sera from doubly vaccinated HCWs ($n = 20$). Full circles denote healthcare workers vaccinated with BNT162b2 ($n = 11$), whereas open circles denote vaccination with AZD1222 ($n = 9$). Numbers in brackets denote fold changes. Wilcoxon matched pairs signed rank tests was used in panel (**c**), (**f**) and (**g**). Student's test was used in panels (**d**) and (**e**). IC50 was calculated by fitting a non-linear regression curve using Graphpad Prism 8 software. All n values constitute biologically independent samples.

stocks in human ACE2 expressing cells (Supplementary Fig 3). These titrations indicated that, in general, RaTG13's human ACE2 usage was less sensitive to the insertion of RBD mutations than SARS-CoV-2, aside from RaTG13 Y498Q which appeared lethal (Supplementary Fig 3a–d). For SARS-CoV-2 the Q493Y substitution caused the largest drop in infectivity (~2 log[10]).

Lastly, the established importance of SARS-CoV-2 E484K substitutions in immune escape, and the presence of this mutation in circulating VOCs led us to investigate the effect of a similar mutation on RaTG13. Using a set of sera ($n = 20$) from HCWs who had been vaccinated with two doses of either BNT162b2 or AZD1222, we carried out pMN assays on SARS-CoV-2 and RaTG13 pseudotypes (WT and E484K or T484K, respectively). As expected, SARS-CoV-2 E484K neutralisation was significantly decreased when compared to SARS-CoV-2 (5.8 fold change, $p = <0.0001$) (Fig. 2f). Interestingly however, in separate experiments with RaTG13 pseudotypes we observed the inverse trend with the corresponding mutation T484K, which showed a non-significant increase in neutralisation when compared to RaTG13 WT (1.2 fold change, $p = 0.1054$), highlighting that the significance of 484 K changes to virus neutralisation are spike-background specific. Building on these findings, we also examined neutralisation of the RaTG13 T403R Spike mutant, which has previously been shown to increase affinity for human ACE2[14]. Pseudotypes bearing this meeting exhibited a significant decrease in overall neutralisation (1.7 fold change, $p = <0.0001$), identifying a correlation between increased affinity for the human ACE2 receptor and decreased neutralisation.

## Discussion

Within the last twenty years, three outbreaks of pathogenic coronaviruses have been recorded, with the current SARS-CoV-2 outbreak resulting in the global COVID-19 pandemic[15]. Since the SARS-CoV outbreak in 2002-2003, substantial focus has been placed on understanding the epidemiology of coronaviruses in order to assess risk and to prevent future outbreaks. This spurred large ecological surveillance studies, culminating in the discovery of numerous coronaviruses in bats, with one group reporting the detection of 293 coronaviruses from a single cave[3,16], nine of which were identified as betacoronaviruses, with one of these given the denominator RaTG13. Several bat coronaviruses, including RaTG13 were found to use the ACE2 receptor for entry and displayed broad species tropism[17–20]. RaTG13 is one of the closest known relatives of SARS-CoV-2 and binds to the human

ACE2 receptor[11,20]. Our study therefore sought to assess the level of cross-neutralisation of RaTG13 afforded by antibodies raised against SARS-CoV-2 either following natural infection, vaccination or both. Despite considerable variation within the RBD we identified that SARS-CoV-2 specific immune sera generated through infection or immunisation is capable of neutralising RaTG13 pseudotypes. Similar cross-neutralisation has been observed in macaques vaccinated with a multimeric SARS-CoV-2-based RBD vaccine[21], a limited set of SARS-CoV-2 convalescent sera[22], as well as in Covid-vaccinated SARS-CoV-1 and SARS-CoV-2 patients[23] demonstrating the breadth of immunity that can be generated against sarbecovirus Spike proteins. However, these studies did not attempt to characterise the amino acid residues in Spike responsible for this cross-neutralisation. These findings have important implications for our understanding of betacoronavirus immunology and the future emergence of these viruses in humans, where herd immunity levels may be maintained at a high level.

Separately, within the current pandemic there are major concerns that new variants of SARS-CoV-2 could arise that evolve to escape host-derived immunity. Our data shows the substantial reduction in neutralisation by the Beta VOC, B.1.351 (Fig. 1c), which is consistent with others in the field[8,10,13,24] exemplifying the concern regarding immune escape. It is now well established that the E484K change present within the Beta RBD plays an important role in neutralising antibody evasion[25], data which we separately confirmed with a single point mutant (Fig. 2g). Indeed, similar data exist for mutations at position N439[12,26,27], Y449[28], E484[29], F486[30], Q493[31], N501[32–34] and Y505[35] within SARS-CoV-2. These sites also differ between the SARS-CoV-2 and RaTG13 RBD. With this in mind, it was therefore surprising to see that RaTG13 was efficiently neutralised by SARS-CoV-2 specific sera despite so many relevant changes in the RBD. These data indicate that SARS-CoV-2 immunity may tolerate considerable changes to the RBD sequence without losing efficacy, and perhaps more optimistically that only a small number of changes exist which can dramatically alter neutralisation, conclusions which are supported by other recent observations[32]. Collectively, these data demonstrate that there must be antigenic epitopes maintained between SARS-CoV-2 and RaTG13 spike. These epitopes are recognised by SARS-CoV-2 specific sera (convalescent or vaccine-derived); however, these sera do not efficiently neutralise more genetically distant sarbecoviruses such as SARS-CoV-1 or WIV16. Of note, we did not assess the role of the four NTD substitutions in RaTG13 (F32S, S50L, T76I,

Q218P) and these may play a small role in the differential neutralisation observed.

There are various explanations which might explain why RaTG13 is more potently neutralised than SARS-CoV-2. A reduced receptor binding affinity for human ACE2[5], when compared to SARS-CoV-2, might mean the RaTG13 Spike is more easily displaced from its receptor by competition from higher affinity antibodies. Supportive of this hypothesis, the N501D and Q493Y changes in SARS-CoV-2, were shown to reduce particle infectivity in human ACE2 expressing cells (Supplementary Fig 3) yet increased neutralisation of these mutants (Fig. 2f). The decreased neutralisation of the RaTG13 multi RBD mutant which contains various substitutions that should enhance human ACE2 binding is also supportive of this "affinity model". In contrast the T484K change in RaTG13, which we might expect to increase human ACE2 usage, actually increased neutralisation. However, it is likely the broader context for these changes in the overall structure of the RBD, and epitopes within, is also important. To address this hypothesis more specifically we finally introduced the T403R mutation into RaTG13, a substitution which has previously been shown to increase human ACE2 binding affinity[14]. In this context the neutralisation of RaTG13 decreased, relative to WT, again providing support for our affinity-based theory for superior neutralisation of this spike.

Data on SARS-CoV-1 and SARS-CoV-2 support a model that these viruses acquired higher human ACE2 usage during spillover[7,36]. If, as we suspect, lower affinity interactions are more sensitive to cross-neutralisation this could provide evidence for SARS-CoV-2 vaccination as a route to prevent subsequent spillover of related sarbecoviruses. This is assuming that the majority of sarbecoviruses have an inherently lower affinity for the human ACE2 receptor whilst circulating in their natural reservoir. However, a recent study has reported the identification of sarbecoviruses in bats from Laos that have RBD sequences that are almost identical to that of SARS-CoV-2[37]. Whilst these isolates would presumably be neutralised by SARS-CoV-2 specific sera, because of this high homology, the more variable RBD of the Omicron variant[38] demonstrates that there is a relatively high degree of plasticity for maintaining efficient use of human ACE2.

In summary, our data show that RaTG13 is potently neutralised by antibodies in convalescent sera from SARS-CoV-2 previously infected and/or vaccinated patients, suggesting that future potential spillover of RaTG13 or a closely related virus may be mitigated by pre-existing immunity to SARS-CoV-2 within the human population. How far this umbrella of cross-neutralisation extends across more distantly related sarbecoviruses is the source of continued research in our laboratory; however, a recently published study from Tan et al., indicates this could include SARS-CoV-1 (Tan et al., 2021). Furthermore, the efficient neutralisation of RaTG13, despite its large number of RBD substitutions highlights that variation within the SARS-CoV-2 Spike may be, to a certain degree, controllable by existing vaccines and/or VOC-based boosters. Emerging data on neutralisation of Omicron, which has >30 amino acid substitutions in Spike, is supportive of this hypothesis[39]. Ultimately, our results suggest that the current priority should remain the effective identification and sequencing of SARS-CoV-2 variants, as it is these viruses which contain the most potent neutralisation-escape mutations.

## Methods

**Tissue culture**. Human Embryonic Kidney 293 T/17 (HEK293T/17) cells were cultured using Dulbecco's modified eagle medium (DMEM, PanBiotech)

supplemented with 10% foetal bovine serum (PanBiotech) and 1% penicillin/streptomycin (PanBiotech), in a 37 °C, 5% $CO_2$ incubator. Cells were routinely passaged three times a week to prevent overconfluency.

**Plasmid generation**. The RaTG13 construct, the chimeric expression plasmids expressing SARS-CoV-2 with the RaTG13 RBD and RaTG13 with the SARS-CoV-2 RBD as well as the individual mutants were synthesised commercially (BioBasic) and subcloned into pcDNA3.1+ expression vectors, as detailed in Conceicao et al.[7]. The B.1.351 (Beta) variant Spike was synthesised commercially (GeneArt) and subcloned into a pCAGGS expression vector. The SARS-CoV-2 Spike expression plasmid was kindly gifted by Professor Xiao-Ning Xu, Imperial College, London.

**Pseudotype virus generation**. The day prior to generating pseudotyped viruses (PV) bearing the Spike protein of either SARS-CoV-2, SARS-CoV-1, WIV16, B.1.351 or RaTG13 and mutants, cells were seeded in T-75 flasks for next day transfection at a density of 70% confluency. On the day of transfection, a DNA mix containing 1000 ng of p.891 HIV Gag-Pol, 1500 ng of pCSFLW luciferase and 1000 ng of either SARS-CoV-2 Spike, RaTG13 Spike, Spike Multi RBD or B.1.351 (Beta) variant were prepared in 200 μL of OptiMEM, followed by addition of transfection reagent FuGENE HD (Promega) at a 1:3 ratio and incubated for 15 min. During this time, culture media was replenished, and transfection complexes were added dropwise into the culture flasks. PVs were harvested 48 hours post-transfection by filtering culture media through a 0.45 μm cellulose acetate filter. PVs were then titrated and aliquoted for storage at −70 °C[40].

**Pseudotype virus titration**. Target cells were prepared the day prior to titration of PVs by transfecting HEK203T/17 cells with 2000ng ACE2 and 150 ng of TMPRSS2 plasmids using FuGENE HD, to render cells permissible to SARS-CoV-2 and RaTG13 PVs. To titrate PVs, 100 μL of undiluted PVs were serially diluted 2-fold down a white 96-well F-bottom plates (Perkin Elmer) in 50 μL of DMEM. Target cells were added at a density of 10,000 cells were per well, and plates were returned to the incubator for 48 h prior to lysis using Bright-Glo reagent (Promega). Luminescence was measured using a GloMax luminometer (Promega). PVs were quantified based on relative luminescence units per ml (RLU/ml)[40].

**Neutralisation assays**. Briefly, heat inactivated human convalescent sera was diluted 1:40 with DMEM and serially diluted 2-fold down white 96-well F-bottom plates. PVs were added at a density of $5 \times 10^5$ RLU/ml in each well and incubated for 1 h at 37 °C and 5% $CO_2$, prior to the addition of target cells at a density of 10,000 cells per well. Plates were returned to the incubator for 48 hours prior to lysis with Bright-Glo reagent. Luminescence was measured using a GloMax luminometer (Promega)[40]. Data were analysed using GraphPad Prism software to derive $IC_{50}$ values[41].

**Serum sample collection**. Serum samples were obtained from healthcare workers (HCW) working at Royal Papworth Hospital, Cambridge, UK (RPH) and from COVID-19 patients referred to RPH for critical care during the first wave of the SARS-CoV-2 pandemic in the United Kingdom (Study approved by Research Ethics Committee Wales, IRAS: 96194 12/WA/0148. Amendment 5). HCWs from RPH were recruited through staff email over the course of 2 months (20th April 2020–10th June 2020) as part of a prospective study to establish seroprevalence and immune correlates of protective immunity to SARS-CoV-2[42]. Patients were recruited in convalescence either pre-discharge or at the first post-discharge clinical review. Patient sera (n = 20) and seropositive HCW sera (n = 5) were obtained between 6th of June 2020 and 22nd of September 2020. Sera samples from HCW immunised with single (n = 21) and double doses (n = 20) of either Pfizer BNT162b2 (1st dose: n = 12, 2nd dose: n = 12) or AZD1222 (1st dose: n = 9, 2nd dose: n = 11) vaccines were obtained 4–6 weeks after each dose of vaccination. All participants provided written, informed consent prior to enrolment in the study.

At recruitment, HCW were classified as pre-exposed according to the results provided by a CE-validated Luminex assay detecting N-, RBD- and S-specific IgG, a lateral flow diagnostic test (IgG/IgM) and an Electro-chemiluminescence assay (ECLIA) detecting N- and S-specific IgG as previously described in ref. [42]. Any sample that produced a positive result by any of these assays was classified as positive.

**Statistics and reproducibility**. All statistical analyses between datasets in this manuscript were performed using GraphPad Prism version 8.0.2. The number of independent biological samples and type of test are described in each figure legend.

**Reporting summary**. Further information on research design is available in the Nature Research Reporting Summary linked to this article.

## Data availability

The datasets generated during and/or analysed during the current study are available from the corresponding author on reasonable request. The source data for each figure is provided in Supplementary Data 1.

## Code availability
No custom code was used in the analyses provided.

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

## Acknowledgements
D.B., J.N., N.T. and A.M. were funded by The Pirbright Institute's BBSRC institute strategic programme grant (BBS/E/I/00007038) and by the MRC funded grant G2P-UK; A National Virology Consortium to address phenotypic consequences of SARS-CoV-2 genomic variation, (MR/W005611/1). D.C., M.M.N., A.N., A.C., P.S., J.C.O., H.B., N.J.T. and J.L.H. are members of Humoral Immune Correlates to COVID-19 (HICC) consortium, funded by the UKRI and NIHR;(COV0170 - HICC: Humoral Immune Correlates for COVID19, Grant Reference code: MC_PC_20016). J.L.H. & S.V. are funded by Innovate UK 72845 DIOS-CoVax and CEPI. MMN and NJT are funded by Wellcome Trust/UK FCDO (GB-CHC-210183). We thank the RPH Foundation Trust COVID-19 Research and Clinical teams, HCWs and Outpatients who participated in studies undertaken by the HICC.

## Author contributions
D.C., N.J.T. and D.B. conceived and designed the study. D.C. and D.B. wrote the manuscript. N.J.T., J.L.H., J.C.O., H.B., B.C. and D.B. supervised the project. D.C., M.M.N., N.T., A.E. and J.N. carried out the experiments. A.N., A.C. and P.S. processed and provided convalescent sera. S.V. contributed with bioinformatical input regarding point mutations. All authors interpreted the data and provided significant feedback. All authors reviewed the manuscript and approved the final paper.

## Competing interests
The authors declare no competing interests.
