## [Peer Review File · Communications Biology]

Reviewers' comments:

Reviewer#1 (Remarks to the Author):

This is a brief manuscript describing the neutralization activity to pseudovirus which carries the SARS-CoV-2 or RaTG13 spike by SARS-CoV-2 convalescent sera from previously infected patients as well as vaccinated healthcare workers. The authors claimed that the serum neutralization activity to TG13 is higher than SARS-CoV-2. In addition, authors did the neutralization assay using pseudoviruses carrying spike chimeras and mutants of single amino acid substitutions within the RBD and found that most of mutations are tolerant to the neutralization assay. The authors also found that the mutation 484K into TG13 resulted in increased neutralisation, in contrast to the same mutation in SARS-CoV-2, in which the E484K is an immune-escape mutation. Although this is the first report regarding the serological cross-neutralization between the SARS-CoV-2 and TG13, the conclusion is not solid based on the current data and may mislead the authorship.

Comments

1. The major flaw affecting the conclusion is the entry efficiency of TG13. As previously reported by several publications (ref. 5, 6, 14, 30, and others), TG13 spike has a low binding affinity to human ACE2 and entry efficiency in human ACE2 expressing cells. It's not logical to compare the neutralization activity of the two pseudoviruses which have distinct different entry efficiency in human ACE2 expressing cells. It's not unexpected that the serum with high neutralization titer to SARS-CoV-2 can neutralize a distantly related SARS-related CoV and one with low entry efficiency.
2. All neutralization assay is lacking the negative control (buffer and serum from healthy persons without being infected or vaccinated by SARS-CoV-2).
3. This reviewer suggests authors prepare TG13 spike or pseudovirus immunized sera and perform the two way serological cross-neutralization assay.

Reviewer #2 (Remarks to the Author):

In this manuscript, using convalescent sera, the authors tested neutralisation of pseudotyped RaTG13 virus which is a bat derived coronavirus closely related to SARS-CoV-2. They also tested the cross mutant pseudotyped viruses of RaTG13 and SARS-CoV-2 including various sites in RBD using from previously infected patients and vaccinated healthcare workers with two different vaccines. Efficiently neutralizing RaTG13 more than SARS-CoV-2 was found and important amino acid sites affecting neutralization were identified. The author also discusses the causes of different neutralization effects. These results provide information for understanding the cross neutralization of two strains of sarbecovirus. The manuscript has been written well, several concerns are as following.

1. Materials and methods, the serums of infected patient was used in the experiment, however the information of serum was not described enough. Were these serum infected people from the same SARS-CoV-2 strain? If it comes from different strains, is there any difference in RaTG13 neutralization efficiency?
2. The authors tested serum from two vaccines, the types of vaccines need to be briefly introduced. The same question is whether the neutralization effect of sera from different vaccines on various viral strains show statistically divergence?
3. The authors used chimeric Spikes to describe the pseudotyped mutants, In this study, it seems to refer to single site mutation, it's better not to describe them using chimera.

Reviewer #3 (Remarks to the Author):

The origin of SARS-CoV-2 is likely in (horseshoe) bats since closely-related coronaviruses have been detected in various species of Asian horseshoe bats. RaTG13 denotes a bat coronavirus whose genomic information was found in a horseshoe bat from Yunnan province in China and it

shares high genome identity with SARS-CoV-2, especially within the spike (S) protein. However, SARS-CoV-2 and RaTG13 S differ in several amino acid residues within the so-called receptor binding domain (RBD), which interacts the cellular receptor molecule ACE2 and represents the main target for neutralising antibodies. In this study, Cantoni and colleagues investigated cross-neutralisation of RaTG13 S by sera from convalescent COVID-19 patients and healthcare workers vaccinated with Pfizer's or AstraZeneca's COVID-19 vaccine. The authors utilised pseudotyped viral particles (pseudoviruses), bearing wildtype or mutant versions of the S proteins of either SARS-CoV-2 or RaTG13, which constitutes a reliable strategy for this kind of analysis. Their main findings are: (i) antibodies in sera from people recovered from SARS-CoV-2 infection or vaccinated against COVID-19 cross-neutralise RaTG13 S and most; (ii) neutralisation of RaTG13 S is more potent as compared to SARS-CoV-2 S; (iii) exchange of the respective RBDs between SARS-CoV-2 and RaTG13 S partially reverts the neutralisation phenotype; (iv) RBD mutation E484K, which reduces neutralisation sensitivity in the context of SARS-CoV-2 S increases neutralisation sensitivity when introduced in RaTG13 S.

Altogether, this is a clear and well-written study that reports interesting results. The experiments are well conducted and the results support the author's conclusions. Taking a few points (see below) into account, this manuscript can be endorsed for publication.

Specific points:

- It might be more appropriate to show the median instead of mean for the neutralisation data.
- Why were only sera from single vaccinated individuals tested. Sera from fully vaccinated (two shots of either BNT162b2 or AZD1222) should also be analyzed.
- It seems like data for RaTG13 S Y493Q are missing from figure 2E.
- The different mutant S proteins should also be investigated for cell entry efficiency. Do the mutations that increase neutralisation of SARS-CoV-2 S reduce cell entry (and maybe ACE2 affinity)? The results might strengthen the authors' hypothesis that reduced affinity of RaTG13 S for human ACE2 may cause that RaTG13 S "is more easily displaced from its receptor by competition from higher affinity antibodies".
- The authors should also discuss the possibility that antibodies targeting the NTD (N-terminal domain) of the S protein may play a role in the differences between neutralisation of SARS-CoV-2 and RaTG13-S (based on the scheme in figure 2A there are four differences between SARS-CoV-2 S and RaTG13 S).

Dr. Dalan Bailey
Group Leader
The Pirbright Institute
Ash Rd
Guildford
Surrey
UK

4th January 2022

Re: Revised submission of COMMSBIO-21-2339-T “Pseudotyped Bat Coronavirus RaTG13 is efficiently neutralised by convalescent sera from SARS-CoV-2 infected Patients”

Dear Reviewers,

Thank you for taking the time to review our original submission. Below, we provide a point-by-point reply to each of your comments. We have also attached a tracked changes version of the manuscript to our submission.

Reviewer #1:

1. The major flaw affecting the conclusion is the entry efficiency of TG13. As previously reported by several publications (ref. 5, 6, 14, 30, and others), TG13 spike has a low binding affinity to human ACE2 and entry efficiency in human ACE2 expressing cells. It's not logical to compare the neutralization activity of the two pseudoviruses which have distinct different entry efficiency in human ACE2 expressing cells. It's not unexpected that the serum with high neutralization titer to SARS-CoV-2 can neutralize a distantly related SARS-related CoV and one with low entry efficiency.

We politely disagree with the reviewer's conclusions; we found the results to be somewhat surprising. We were expecting to see very low neutralisation of RaTG13, assuming it would be antigenically distinct from SARS-CoV-2 because of the sheer number of RBD substitutions. Identifying superior neutralisation led us to the deeper investigation of the balance between affinity and conservation of antigenicity described in this paper. Since submission, others have published similar findings on RaTG13 neutralisation in Cell, Nature and NEJM, albeit with less controls, less mutational analysis and without detailed measures of particle infectivity (Tan et al., Saunders et al., Liu et al., all 2021). To complement these findings, we have now added data on more genetically and antigenically distinct beta coronaviruses SARS-CoV-2 and WIV16 (revised Figure 1B). Like RaTG13, WIV16 has lower affinity for human ACE2 and is distantly related, but this pseudovirus was not neutralised efficiently by SARS-CoV-2 specific sera. In addition, we also added data on a higher affinity mutant of RaTG13 (T403R; Figure 2G) which showed lower neutralisation IC50s, data which fits with the hypotheses outlined in the discussion. Collectively, we hope these data provide more context for our comparisons of SARS-CoV-2 and RaTG13 neutralisation. We would also highlight that entry and ACE2 usage are distinct from antigenicity; in our assays neutralisation is calculated by comparison to the RLU recorded for that individual pseudotype, i.e. pseudotype plus

target cells with no sera, so each IC50 is relative and takes into account differences in infectivity. To provide more information on how this was assessed we have also added data on the different pseudoparticle's relative infectivity (Supplemental Figure 3). These areas of discussion have also been added throughout the manuscript.

2. All neutralization assay is lacking the negative control (buffer and serum from healthy persons without being infected or vaccinated by SARS-CoV-2).

These controls have now been added to the manuscript in Supplemental Figure 1. This includes buffer alone (no sera) and negative control sera, as well as exemplar neutralisation data for the neutralisations presented in Figure 1B. See lines 164-169 of revised manuscript.

3. This reviewer suggests authors prepare TG13 spike or pseudovirus immunized sera and perform the two way serological cross-neutralization assay.

Unfortunately, we do not have the facilities or the relevant UK animal licence to do these experiments.

Reviewer #2:

1. Materials and methods, the serums of infected patient was used in the experiment, however the information of serum was not described enough. Were these serum infected people from the same SARS-CoV-2 strain? If it comes from different strains, is there any difference in RaTG13 neutralization efficiency?

Unfortunately, we do not have confirmed information on which virus variant these people were infected with; however, the time of sampling (April 2020-June 2020) would indicate these are most likely from the first wave and likely Wuhan D614G/ B.1. We have indicated this in the results section of the manuscript (lines 159-160).

2. The authors tested serum from two vaccines, the types of vaccines need to be briefly introduced. The same question is whether the neutralization effect of sera from different vaccines on various viral strains show statistically divergence?

We have added a more detailed description of the vaccines to the main text, as follows, "BNT162b2 (Pfizer–BioNTech) is a lipid nanoparticle–formulated, nucleoside-modified RNA vaccine encoding prefusion stabilized, SARS-CoV-2 spike while ChAdOx1 nCoV-19 (AZD1222, Oxford–AstraZeneca) is an adenoviral-vectored vaccine expressing wild type (non-stabilised) spike" (lines 172-175). We also performed the statistical analysis requested and have included this in Supplemental Figure 2 together with text in the manuscript (lines 183-185).

3. The authors used chimeric Spikes to describe the pseudotyped mutants, In this study, it seems to refer to single site mutation, it's better not to describe them using chimera.

We apologise for the confusion. The chimeras we described are indeed multi-site mutants containing all the single mutants listed in 2D and 2E. To clarify this, and remove the chimera term, we have changed the phraseology to "multi-site mutants" throughout the manuscript and figures.

Reviewer #3

1. It might be more appropriate to show the median instead of mean for the neutralisation data.

The medians are now shown.

2. Why were only sera from single vaccinated individuals tested. Sera from fully vaccinated (two shots of either BNT162b2 or AZD1222) should also be analyzed.

This was the only sera available at the time; however, sera from double vaccinees was subsequently used in 2F and 2G and showed similar results, although interestingly boosting or infection reduced the enhanced neutralisation of RaTG13 somewhat (from 4.5x to 1.2x in Figure 1D). Indeed, in this study, and other we are involved with, we have found that a single dose is more sensitive for picking up antigenic differences between viruses. As such we believe it's use is warranted in this context.

3. It seems like data for RaTG13 S Y493Q are missing from figure 2E.

The RaTG13 Y498Q mutant couldn't be rescued to high titres. We have added reference to this in the text (lines 213-220) and an asterisks to the row title in the figure (Figure 2D). Also, data on infectivity of this virus, or lack thereof, has been added to Supplemental Figure 3.

4. The different mutant S proteins should also be investigated for cell entry efficiency. Do the mutations that increase neutralisation of SARS-CoV-2 S reduce cell entry (and maybe ACE2 affinity)? The results might strengthen the authors' hypothesis that reduced affinity of RaTG13 S for human ACE2 may cause that RaTG13 S "is more easily displaced from its receptor by competition from higher affinity antibodies".

We would like to thank the reviewer for this suggestion. These data have been added to the Supplemental Figure 3, and they do indeed support this hypothesis, with Q493Y and N501D having lower infectivity but increased neutralisation. We have modified the discussion to reflect this added data (lines 282-285).

5. The authors should also discuss the possibility that antibodies targeting the NTD (N-terminal domain) of the S protein may play a role in the differences between neutralisation of SARS-CoV-2 and RaTG13-S (based on the scheme in figure 2A there are four differences between SARS-CoV-2 S and RaTG13 S).

We agree with the reviewer on this point and a relevant line of discussion has been added to the main text, "Of note, we did not assess the role of the four NTD substitutions in RaTG13 (F32S, S50L, T76I, Q218P) and these may play a small role in the differential neutralisation observed" on lines 277-278.

Yours sincerely,
Dalan Bailey
On behalf of all co-authors.

Dr. Dalan Bailey

REVIEWERS' COMMENTS:

Reviewer #1 (Remarks to the Author):

This reviewer has no more comments.

Reviewer #2 (Remarks to the Author):

The authors answered all the concerned, no further questions.

Reviewer #3 (Remarks to the Author):

The authors addressed all my points and provided additional data and/or reasonable explanation.
The revised manuscript is now suitable for publication.